# Seasonal Variation and Vertical Distribution of Inorganic Nutrients in a Small Artificial Lake, Lake Bulan, in Mongolia

Ariunsanaa Baterdene [1,2,3], Seiya Nagao [4,*], Baasanjav Zorigt [2], Altansukh Ochir [2,3], Keisuke Fukushi [4], Davaadorj Davaasuren [5], Baasansuren Gankhurel [6], Enkhuur Munkhsuld [3,7], Solongo Tsetsgee [3] and Ariuntungalag Yunden [2]

1   Division of Material Chemistry, Graduate School of Natural Science and Technology, Kanazawa University, Kanazawa 920-1192, Ishikawa, Japan; ariunsanaa1790@stu.kanazawa-u.ac.jp
2   Department of Environment and Forest Engineering, School of Engineering and Applied Sciences, National University of Mongolia, Ulaanbaatar 14201, Mongolia; baaska8696@gmail.com (B.Z.); altansukh@num.edu.mn (A.O.); ariuna_0924@yahoo.com (A.Y.)
3   Institute for Sustainable Development, National University of Mongolia, Ulaanbaatar 210646, Mongolia; munkhsuld.e.aa@m.titech.ac.jp (E.M.); osko_8888@yahoo.com (S.T.)
4   Institute of Nature and Environmental Technology, Kanazawa University, Kanazawa 920-1192, Ishikawa, Japan; fukushi@staff.kanazawa-u.ac.jp
5   Department of Geography, School of Art and Sciences, National University of Mongolia, Ulaanbaatar 210646, Mongolia; davaadorj@num.edu.mn
6   Division of Global Environmental Science and Engineering, Graduate School of Natural Science and Technology, Kanazawa University, Kanazawa 920-1192, Ishikawa, Japan; gbaasnsrn@stu.kanazawa-u.ac.jp
7   Department of Transdisciplinary Science and Engineering, School of Environment and Society, Tokyo Institute of Technology, Yokohama 226-8502, Kanagawa, Japan
*   Correspondence: seiya-nagao@se.kanazawa-u.ac.jp; Tel.: +81-761-51-4440

**Abstract:** This is the first seasonal observation study on nutrient dynamics undertaken in a small freshwater lake with eutrophication in Mongolia. The vertical profile and seasonal fluctuation of nutrients are crucial to understanding the biogeochemical cycles in aquatic systems. In this study, field research was carried out at a small and shallow lake, Lake Bulan, in the lower Kharaa River basin. The area has been receiving increased nutrient loads from the water catchment area for the last 20 years. Water samples were collected seasonally from the lake from 2019–2022 and analyzed for nutrients, major cations, trace metals, and dissolved organic carbon. The average concentration of dissolved inorganic nitrogen (DIN) in the surface lake water had a wide seasonal variation from $0.26 \pm 0.11$ mg N/L in August to $1.44 \pm 0.08$ mg N/L in January. Seasonal differences were also observed in the vertical profiles. Concentrations were relatively similar at the various water depths in April and September at turnover time. Thermal stratification was observed when the lake was covered in ice, with the maximum concentrations being observed in the bottom layer in the months of January and August. The phosphate concentration showed a similar variation trend. These results indicate that both the summer and winter stratifications are important for regeneration of nutrients in the bottom layer, biochemical cycling, and mitigating impacts of global warming on small and shallow lakes in Mongolia.

**Keywords:** nutrient dynamics; artificial freshwater lake; vertical water profile; Lake Bulan

## 1. Introduction

Eutrophication of freshwater lakes has become a serious environmental problem due to anthropogenic activity and climate change [1,2]. According to the Water Research Commission, 28 to 54% of lakes globally are facing eutrophication problems such as enriched dissolved nutrients, phytoplankton overgrowth, decline in dissolved oxygen, and fish-mortality [3]. Eutrophication, as a result of industrial, agricultural, and other activities, is considered to be the most notable negative impact on water bodies [4,5]. These

anthropogenic impacts can cause water bodies with low nutrient levels to transition to a eutrophic state resulting in dramatically reduced ecosystem services [4]. The effects of climate change on freshwater flow and rates of nutrient cycling are resulting in increased eutrophication due to the increase in temperature and change in precipitation patterns.

Shallow lakes are more common than deep lakes and provide many ecosystem services including subsistence, drinking water, irrigation water, pollution dilution, transportation, recreation, and aesthetic enjoyment [5]. Shallow lakes are considerably more vulnerable than deep-water ecosystems because they have a low capacity for diluting contaminants and nutrient loads [6], limited oxygen storage capacity, and reduced oxygen availability due to decomposition of organic matter [7]. Small shallow lakes are also more vulnerable as there is lower inflow and outflow, which affects the level of water mixing and vertical transportation of settling particles, thereby influencing pollution dispersion capability [8].

Small lakes at temperate latitudes provide important indicators of the impacts of anthropogenic activity and global warming on water quality, which include the duration of thermal stratification in summer and lake mixing regime [9,10]. Stable stratification prevents the exchange of water between the surface and bottom layers and has a significant impact on the physical and chemical parameters of the water quality [10]. Thermal stratification obstructs the reaeration pathway to the bottom layer, and when dissolved oxygen (DO) in the water is consumed during organic matter decomposition, it results in the formation of an anaerobic environment in this layer [11,12]. Under anoxic conditions, organic matter mineralization promotes the release of ammonium nitrogen and phosphorus from sediments [13]. The changes in lake mixing regimes result in changes to nutrient dynamics and ultimately lake productivity [14]. The speed of DO depletion in the hypolimnion of eutrophic lakes caused by summer stratification is increased due to the earlier onset of thermal stratification combined with higher vertical stability. These phenomena have been observed in several temperate lakes, including Lake Mendota [15] and Lake Mohonk [4] in the northern part of the USA and Lake Blelham Tarn in the United Kingdom [12].

Interest in winter nutrient concentrations and biogeochemical cycles in lakes under ice cover has grown rapidly in recent years [16–18]. Winter stratification caused by ice cover may affect nutrient availability for phytoplankton in the spring and summer seasons that follow. Nutrients are released into the hypolimnion, resulting in the decomposition of dead organic material in the benthic sediments [19–21]. Additionally, during winter, nutrients are released from water during the freezing process, which leads to an increase in concentration in the water beneath the ice [16]. The findings from the study at the shallow Lake Ulansuhai, in central Mongolia, confirmed that ice cover can dramatically increase nutrient concentrations in water during the winter [22]. Higher concentrations of $PO_4^{3-}-P$ and $NO_3^--N$ were observed in ice-covered water during winter in Lake Arakhley [23] and Lake Gusinoe [24] in the Transbaikal Region of Russia. Powers et al. [25] reported that the nitrate accumulation was strongly correlated to the number of days from the onset of ice cover in the temperate lakes Allequash, Big Muskellunge, Crystal, Sparkling, and Trout in North America.

Vertical profiles of water quality parameters (WQP) and nutrients, and their seasonal variations, are crucial to understanding biogeochemical cycles and nutrient dynamics. In Mongolia, the seasonal thermal regime, distribution of physicochemical parameters, and biogenic substances have primarily been studied in large, deep lakes such as Lake Khuvsgul [26] and Lake Uvs [27], as well as in western saline lakes, Lake Telmen, Lake Oigon, Lake Tsegeen, and Lake Khag [28]. However, even though 85% of lakes in Mongolia are classified as small with a water surface area of less than 1 $km^2$, and considering small lakes and reservoirs are considerably impacted by anthropogenic pollution and nutrient load, they have not yet been investigated [29]. Tao et al. reported that rapid loss of lakes occurred in the Mongolian Plateau (Mongolia, central Mongolia) from 1980–2010 [30]. In particular, the number of small lakes with a water area between 1–10 $km^2$ decreased significantly, with 58 lakes with a combined surface area of 220.2 $km^2$ drying up. This is similar to the trend observed in a study by Davaa [31], who noted that the lake area

in Mongolia was reduced by 373 km$^2$ from 1940s–2000s. Both studies linked this issue to the impact of climate change on precipitation patterns and increase in mean annual temperature [32–35]. It is important to establish the current situation in these lakes for a better understanding of potential future changes in biogenic substances in Mongolian shallow lakes. The aim of our study was to identify the nutrient dynamics and distribution profile in a small shallow eutrophic lake, Lake Bulan, which has a high nutrient load as a result of anthropogenic activity, and is located in the lower part of the Kharaa River basin [36]. Field research was performed seven times over the time period 2019–2022. The nutrient dynamics and changes in the vertical profiles were investigated for each season, including the ice-covered winter period.

## 2. Materials and Methods

Lake Bulan is an artificial freshwater lake formed from open cast sand and gravel mining during the construction of Darkhan city in the 1960s. The biophysical features are as follows: eutrophic green color, permanent water body; surface area of 0.45 km$^2$ [37] and depth of 6–15 m; 1.1 km long and 0.6 km wide; and elevation of 678 m a.s.l. Lake Bulan is a lentic water body without inflow or outflow water courses. The main recharge sources are precipitation and groundwater from the Kharaa River valley [38]. Ice forms on the lake surface at the beginning of November. The lake is poor in terms of floating and bottom species [38]. It is fed by groundwater, and the growth of animals is low, depending on the depth and temperature of the water. Crayfish and leeches have spread to the rocks on the shores of the lake [38]. Lake Bulan is representative of the lakes of the Khangai–Khentei mountainous region, due to its geographical position. This mountainous region is characterized by the predominance of small and medium lakes [29].

To identify temporal changes in nutrient concentration levels and water quality conditions in Lake Bulan, water samples were collected in winter (3 January 2020 and 30 January 2021), spring (30 April 2020), summer (27 August 2020), and autumn (24 September 2021). Surface and bottom water samples were collected from two stations (Bulan 1 and Bulan 3) (Figure 1). To evaluate the vertical distribution of nutrients, major cations, and trace metals, samples were also collected from the Bulan 2 station at depth intervals of 1 m. Water quality parameters (WQP) were measured in the field using a portable multi-parameter water quality meter WQC-24 (TOA-DKK Corp., Tokyo, Japan) on 24 September 2021 (autumn), 25 February 2022 (winter), and 30 April 2022 (spring). In summer, vertical WQPs were measured using a portable multi-parameter meter WN-32EP and DO-31P (TOA-DKK Corp., Tokyo, Japan). Water samples were obtained using a transfer pump GEO-pump-CFC-a (Geo Science Laboratory Ltd., Kanagawa, Japan).

To compare nutrient concentrations in Lake Bulan with other small freshwater lakes, surface water samples were collected from Lake Bornuur on 26 September 2019 and from Lake Ugii on 22 August 2020 (Table 1, Figure 2). Lake Bornuur is a small artificial lake in the upper catchment area of the Kharaa River basin and was formed between 1968–1970. Lake Ugii is a natural freshwater lake with drainage to the Orkhon River on the eastern edge of the Khangai Mountains [39].

**Table 1.** Sampling sites and their locations in Lake Bulan, Lake Ugii, and Lake Bornuur.

| Station Name | Coordinate | | Description |
| --- | --- | --- | --- |
| | N | E | |
| Bulan 1 | 49°31′5″ | 105°54′49″ | 7 m water depth |
| Bulan 2 | 49°30′52″ | 105°54′47″ | 9 m water depth |
| Bulan 3 | 49°30′42″ | 105°54′47″ | 7 m water depth |
| Bornuur | 48°26′48″ | 106°13′8″ | Upper reaches of Kharaa River basin, area 0.25 km$^2$, 4.6 m water depth |
| Ugii | 47°46′17.35″ | 107°46′29.89″ | Khangai Mountains, area 25.7 km$^2$, 15 m water depth |

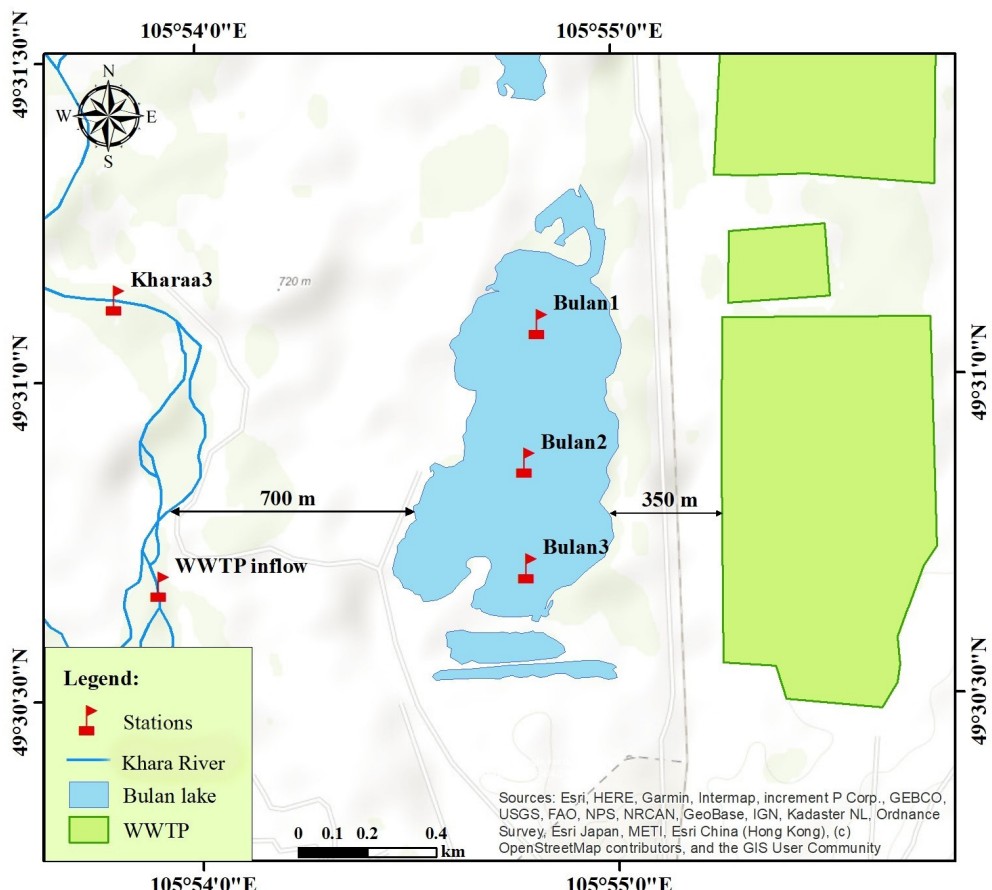

**Figure 1.** Research area: stations at Lake Bulan. Map cited from Sources: Esri. HERE. Garmin. Intermap. Increment P Corp., GEBCO, USGS, FAO. NPS, NRCAN. GeoBase. IGN. Kadaster NL. Ordnance Survey. Esri Japan, METI, Esri China (Hong Kong), Open Street map contributors, and the GIS User Community and is modified.

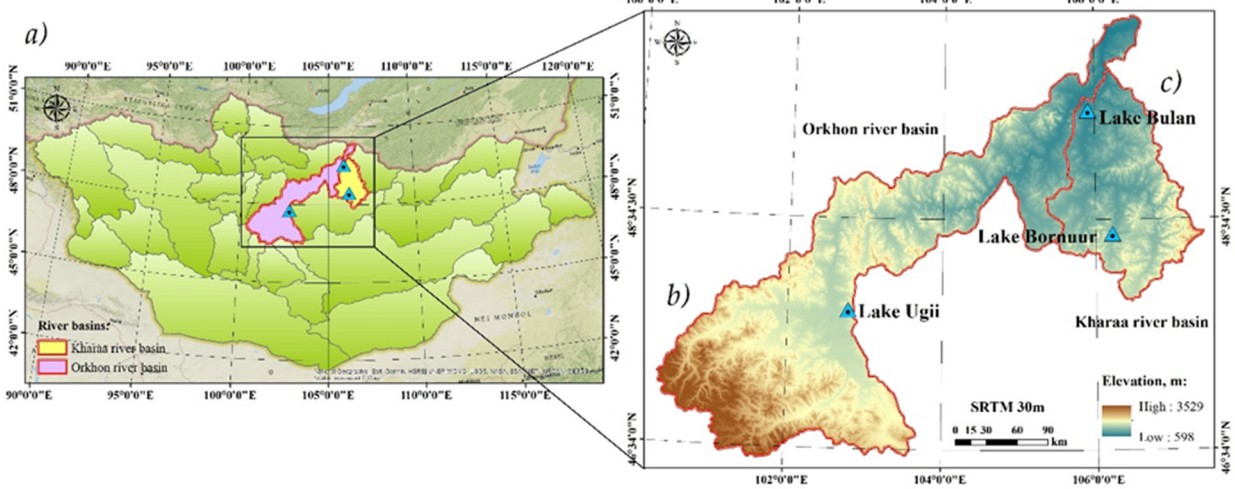

**Figure 2.** Research area: (**a**) River basins in Mongolia; (**b**) Orkhon River basin; and (**c**) Kharaa River basin. The map cited from Environmental Information Center (EIC) meta data [40], WGS84: 49N, Map cited from Sources: National Geographic, Esri, Garmin, HERE, UNEP-WCMC, USGS, NASA, ESA, METI, NRCAN, GEBCO, NOAA, increment P Corp. and is modified.

A small boat and a sampling pumper were used to collect samples during the seasons without ice cover. Winter samples were collected using an ice drill and the sampling pumper. The thickness of the ice cover on Lake Bulan was approximately 0.9–1.0 m. The spring and summer samples were collected in two 50 mL polypropylene bottles in the field after filtering the water through a 0.45 μm membrane filter. The winter water samples were collected in a 1000 mL plastic bottle, filtered through a 0.45 μm membrane filter, and separated into two 50 mL polypropylene bottles. One sample was frozen at −20 °C for nutrient and dissolved organic carbon (DOC) analyses. The second sample was preserved by adding 0.5 mL of ultrapure nitric acid (60%). The concentrations of inorganic nitrogen ($NH_4^+-N$, $NO_2^--N$, $NO_3^--N$), phosphorus ($PO_4^{3-}-P$), and silicate ($SiO_2-Si$) were analyzed using the segmented flow analyzer QuAAtro39 (Seal Analytical, Mequon, WI, USA). The DOC concentrations were measured using a TOC analyzer TOC-V SCN (Shimadzu Corp., Kyoto, Japan). The concentrations of major cations ($Na^+$, $K^+$, $Ca^{2+}$, and $Mg^{2+}$), trace metals (Fe and Mn), and total phosphorus (TP) were measured using inductive coupled plasma optical emission spectrometry Varian-710 (Varian Inc., Palo Alto, CA, USA). Statistical analysis was provided by Originlab, 2021 version (OriginLab corporation, Northampton, MA, USA).

## 3. Results

### 3.1. Physical and Chemical Characteristics of Lake Bulan

Water quality characteristics and seasonal variation of surface water at the Bulan 2 station are presented in Table 2. The pH ranged from 6.99–9.06, water temperature (WT) variation was 0.1–20.1 °C, and electrical conductivity (EC) was 72.2–100 mS/m. The DO concentration ranged from 2.78–16.8 mg/L, which, barring the spring measurement, was lower than the national standard MNS 4586-98 [41].

**Table 2.** Water quality characteristics of surface lake water at Bulan 2 station.

| Parameter | 30 April 2022 | 27 August 2020 | 24 September 2021 | 25 February 2022 |
|---|---|---|---|---|
| WT, °C | 8.03 * | 20.1 | 14.8 | 0.1 |
| pH | 8.08 | 9.06 | 8.23 | 6.99 |
| DO, mg/L | 16.8 | 2.78 | 4.72 | 4.00 |
| EC, mS/m | 84.1 | 76.6 | 72.2 | 100 |
| Turbidity, NTU | 12.1 | - | 15.5 | 4.00 |

Note: (-) no determination, * WT was ~8–10 °C in surface water in May 2018 [38].

The cationic composition in Lake Bulan is shown in Figure 3. The water from Lake Bulan had a high $Na^+$ concentration. Gibb's plot showed that major processes controlling the water chemistry of Lake Bulan were related to evaporation dominance rates (Supplementary Table S1).

The results of the winter WQP measurements at three stations are shown in Figure 4a. Ice cover was approximately 0.88 m and WT, pH, EC, DO and turbidity were 0.13 ± 0.06 °C, 7.07 ± 0.15, 99.9 ± 0.17 mS/m, 2.48 ± 1.32 mg/L, and 3.33 ± 0.61 NTU in lake water under ice (1 m water depth from the ice surface), respectively. The water temperature had an increasing trend corresponding to depth from 0.1 °C at 1 m to 4.2 °C at 8.8 m at the Bulan 2 station. Bulan 1 and Bulan 3 showed a similar profile. These results indicate that winter stratification was formed due to the surface ice cover [42]. The pH ranged from 7.33–5.5 with lower pH observed at greater depth. A decrease was observed in EC at all three stations up to a depth of 6 m, but increased to 104 mS/m at the deepest Bulan 2 sampling point. The DO concentration profile differed between the three stations. Bulan 1 had a low DO concentration with the lowest concentration present in the surface layer. The DO concentration steadily increased with depth at Bulan 3, while at Bulan 2 DO concentration showed a decrease. Turbidity was consistent at Bulan 1 and Bulan 3; however, it increased sharply to the bottom at Bulan 2.

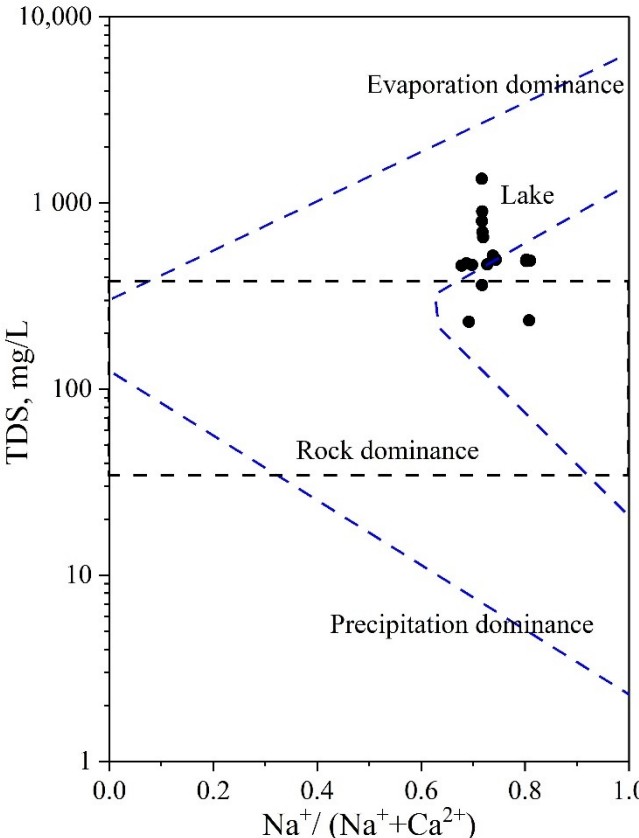

**Figure 3.** Gibbs plot of surface and bottom water samples from Lake Bulan. Note: closed circles—the ratio of cations [(Na $^+$)/(Na$^+$ + Ca$^{2+}$)] against TDS in water Lake Bulan.

Results of the spring WQP measurements are presented in Figure 4b. Consistent with a previous study undertaken at Lake Bulan, WT ranged from 8.03–10.2 °C [38]. The water pH ranged from 8.03–8.73 at the three stations. Surface water EC at the three stations ranged from 80.2–84.1 mS/m. The DO concentration was approximately 14.9–18.35 mg/L in surface water. At station Bulan 1, the highest DO concentration of 18.5 mg/l was observed in the bottom layer. The vertical turbidity profile was similar at the three stations, increasing from the surface to 8 m depth from 12.5 ± 0.4 NTU–16.4 NTU, 15 NTU, and 15.3 NTU in the bottom layers of Bulan 1, Bulan 2, and Bulan 3, respectively.

Summer WQP results are presented in Figure 4c. There was a slight decreasing trend in WT, which was approximately 19–21 °C. The average pH was 9.03 ± 0.09 in the surface layer and decreased steadily to 8.58 in the bottom layer. The EC was 76.7 ± 0.6 mS/m in the surface water and 78.6 ± 2.55 mS/m in the bottom layer. The DO concentration observed was 2.7 ± 0.4 mg/L in surface water and decreased with depth until it reached 1.45 mg/L at a depth of 7 m. The WT, pH and DO concentration profiles indicate that stratification most likely formed at a depth of 4–5 m.

The autumn WQP results are shown in Figure 4d. The average WT, pH, EC, DO, and turbidity values at the three sampling sites were 14.8 ± 0.2 °C, 8.37 ± 0.12, 72.9 ± 1.04 mS/m, 4.41 ± 0.28 mg/L, and 14.7 ± 1.47 NTU, respectively. The WT decreased slightly until reaching a depth of 4 m and then remained constant until reaching a depth of 8 m (maximum). The vertical pH profile was mostly similar at each depth; however, the pH at the Bulan 1 station ranged from 8.00–8.98. The vertical DO concentration profile increased until a depth of 2 m and then decreased thereafter. The vertical turbidity profiles at stations Bulan 2 and Bulan 3 were similar, showing a decreasing trend until reaching a depth of 3 and 4 m, respectively, and remaining constant thereafter until reaching a depth of 8 m (maximum). The turbidity profile trend at Bulan 1 differed in that it increased from 13.0 NTU at the surface to 18.3 NTU at 7 m depth.

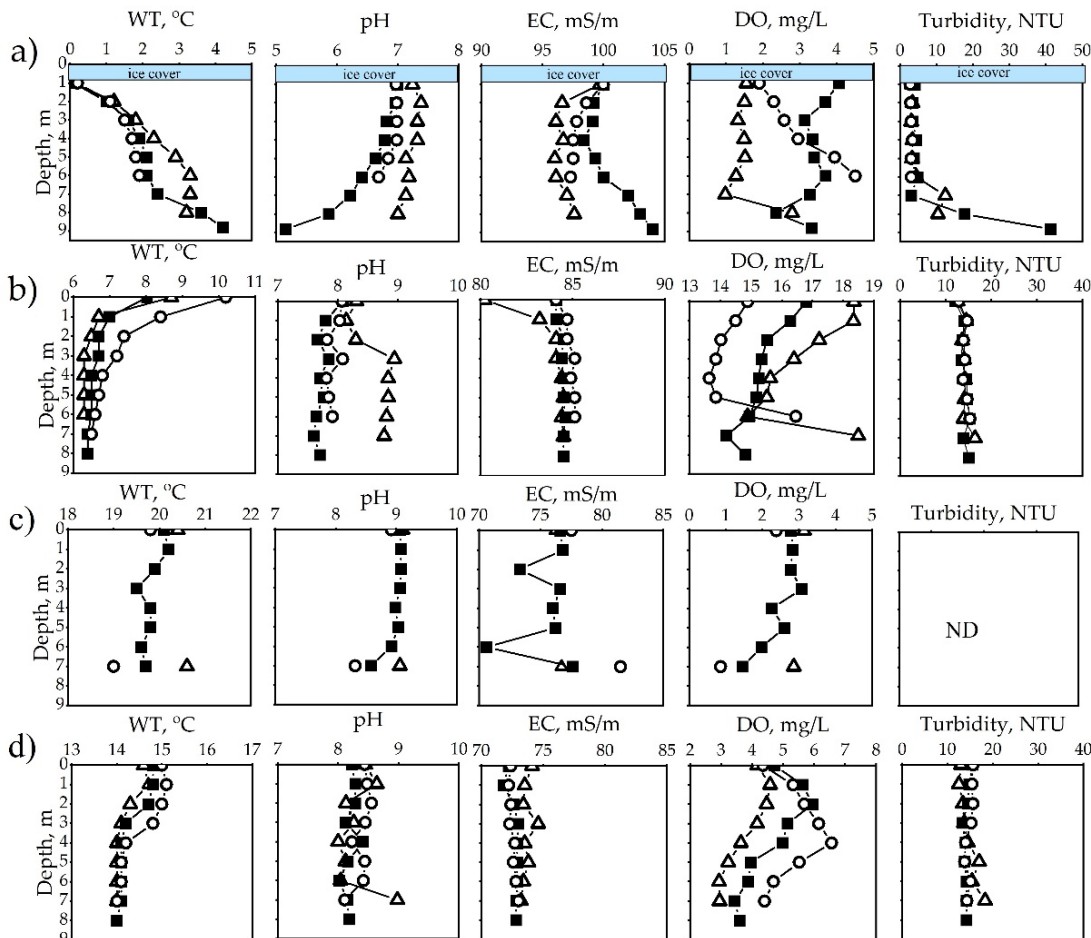

**Figure 4.** Vertical profiles of WQP of Lake Bulan. Note: (**a**) winter (25 February 2022), (**b**) spring (30 April 2022), (**c**) summer (27 August 2020), and (**d**) autumn (26 September 2021); WT—water temperature, EC—electrical conductivity, DO—dissolved oxygen, NTU—nephelometric turbidity unit; open triangle—Bulan 1, closed square—Bulan 2, open circle—Bulan 3; ND—no data.

### 3.2. Variation of Nutrients and DOC Concentration and Their Vertical Profiles

The average concentrations of nutrients and DOC in surface water at three surface stations from Lake Bulan are presented in Table 3. Higher $NH_4^+$, $NO_3^-$, and $PO_4^{3-}$ concentrations were observed in January. The concentration of DOC was higher in the autumn sample.

**Table 3.** Averaged concentrations of nutrients and DOC in surface water from three stations at Lake Bulan.

| Sampling Date | Concentration, mg/L | | | | | |
|---|---|---|---|---|---|---|
| | $NH_4^+$- N | $NO_3^-$ - N | $NO_2^-$-N | $PO_4^{3-}$-P | $SiO_2$-Si | DOC |
| 3 January 2020 | 0.64 ± 0.07 | 0.71 ± 0.11 | 0.02 ± 0.004 | 0.03 ± 0.01 | 1.59 ± 0.83 | 7.81 ± 0.82 |
| 30 January 2021 | 0.49 ± 0.04 | 0.91 ± 0.08 | 0.05 ± 0.005 | 0.09 ± 0.07 | 2.69 ± 0.37 | 6.19 ± 0.28 |
| 30 April 2020 | 0.09 ± 0.02 | 0.68 ± 0.01 | 0.02 ± 0.000 | 0.002 ± 0.001 | 0.85 ± 0.16 | 5.82 ± 0.14 |
| 27 August 2020 | 0.19 ± 0.07 | 0.06 ± 0.035 | 0.005 ± 0.001 | 0.003 ± 0.00 | 1.63 ± 0.09 | 6.19 ± 0.28 |
| 27 September 2019 | 0.25 ± 0.06 | 0.10 ± 0.04 | 0.03 ± 0.01 | 0.034 ± 0.009 | 2.21 ± 0.33 | - |
| 24 September 2021 | 0.48 ± 0.10 | 0.18 ± 0.01 | 0.06 ± 0.002 | 0.016 ± 0.003 | 1.38 ± 0.11 | 9.44 ± 0.68 |

Note: (-) no analyses.

Figure 5 presents vertical profiles of nutrients and DOC concentrations at the three stations. In winter, $NH_4^+-N$, $NO_2^--N$, and $NO_3^--N$ concentrations steadily decreased up to a depth of 5–6 m and then increased sharply beyond that. The concentration of $PO_4^{3-}-P$ increased steadily from a depth of 2 m until the bottom layer (Figure 5a). The higher concentration directly beneath the ice surface was a result of the release of nutrients during ice formation [22]. Spring (Figure 5b) and autumn (Figure 5d) profiles showed relatively constant concentrations in the water column. The summer profile (Figure 5c) showed a sharp increase in $NH_4^+-N$ and $PO_4^{3-}-P$ concentrations with water depth, while $NO_2^--N$ and $NO_3^--N$ concentrations were lowest in the bottom layer.

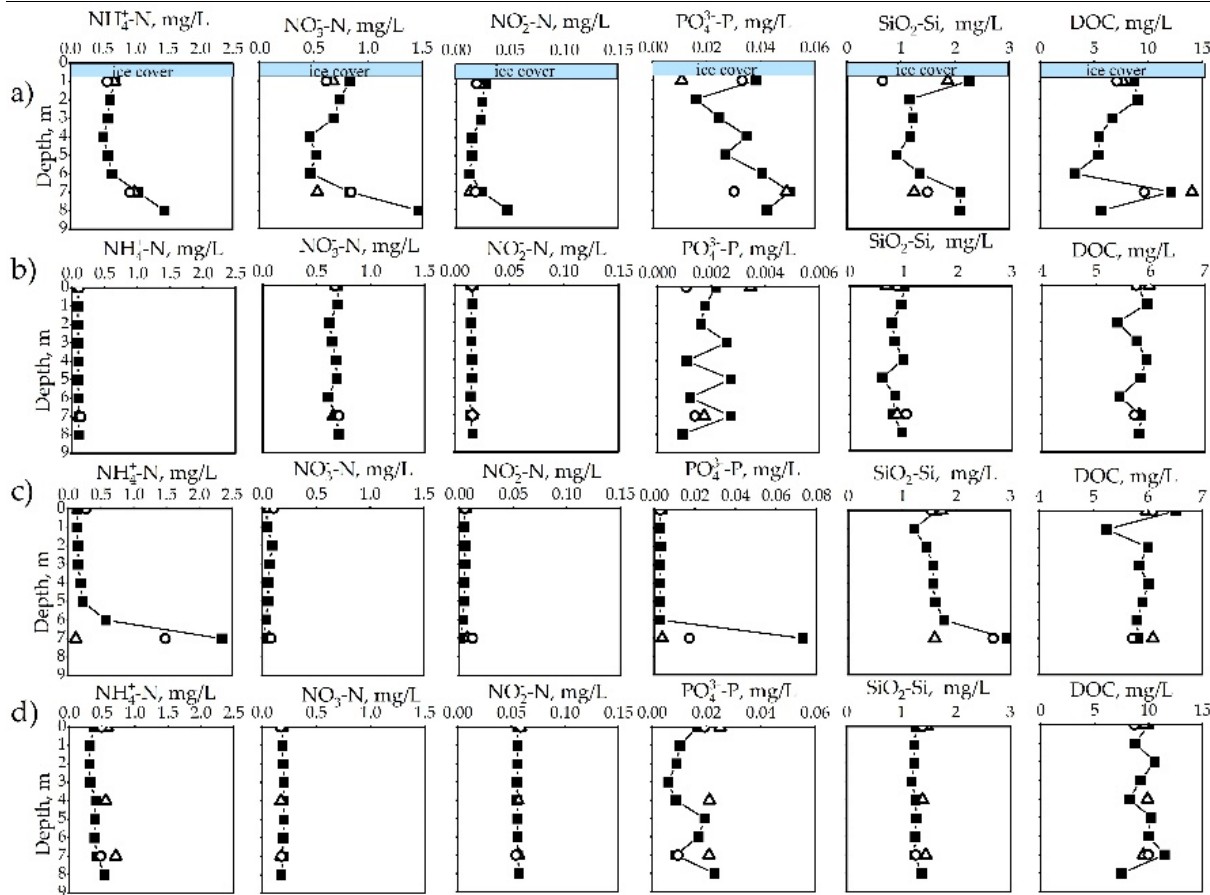

**Figure 5.** Spatial variation and vertical profiles of nutrient and DOC concentrations in Lake Bulan. Note: (**a**) winter (January 2020), (**b**) spring (April 2020), (**c**) summer (August 2020), and (**d**) autumn (September 2021); open triangle—Bulan 1, closed square—Bulan 2, open circle —Bulan 3.

The $NH_4^+-N$ concentration was mostly similar at different depths in April, but in August, it increased substantially from a depth of 5 m, reaching 2.5 mg/L. The January observation results showed an increasing trend with water depth, though the concentrations differed from each other for the bottom layer. The $NO_2^--N$ concentration showed a wide variation from 0.005 mg/L in August to 0.06 mg/L in September. The January samples showed an increase in $NO_2^--N$ concentration from a depth of 6 m to the bottom layer. The vertical profiles of $NO_3^--N$ concentration showed a variation trend similar to the profiles of $NO_2^--N$ concentration. The January profile of $NO_3^--N$ concentration varied relatively widely and decreased until a depth of 6 m, but then increased to 1.46 mg/L in the bottom layer. Phosphate concentration increased in the bottom layer in January, August, and September. The $PO_4^{3-}-P$ concentration increased dramatically and reached its maximum (0.075 mg/L) in the bottom layer in August. The silica concentrations showed a vertical profile similar to that of the phosphate concentrations, with $SiO_2$-Si concentration

fluctuating from 0.5–3.0 mg/L and reaching its maximum in August. There were no notable changes in the DOC concentration profile in April and August. However, the concentration fluctuated from 2.5–12 mg/L during the ice cover period. The DOC concentration was constant over ranging depth in April and August but showed wide variation with a maximum peak at a depth of 7 m in January and September.

We compared the Lake Bulan results to those of other freshwater lakes in similar geographical and meteorological zones in the Mongolian Plateau, Russia, and China. The summarized results are presented in Table 4. Lake Bulan had a relatively high concentration of nutrients—in particular, the dissolved inorganic nitrogen (DIN) concentration (0.77–1.44 mg N/L). The maximum phosphate concentration of Lake Bulan was an order of magnitude higher than in the other Mongolian lakes.

**Table 4.** Comparison of nutrient concentrations in surface water from Lake Bulan and other lakes.

| Country | Lake | Lake Type | Altitude (m) | Area (km²) | Water Depth (m) | | $NH_4^+$ mg N/L | $NO_3^-$ mg N/L | $NO_2^-$ mg N/L | $PO_4^{3-}$ mg P/L | Sampling Time | Reference |
|---|---|---|---|---|---|---|---|---|---|---|---|---|
| Mongolia | Khuvsugul | Deep freshwater lake | 1645 | 2760 | 138 | Seasonal range | 0.006–0.03 | 0.005–0.001 | 0.0001–0.0026 | 0.002–0.006 | 2008–2011 | [26] |
| | Ugii | Shallow freshwater lake | 1332 | 25.7 | 6.6 | August | 0.004 | 0.1 | 0.001 | 0.005 | 2020 | This study |
| | Bornuur | Shallow freshwater | 1007 | 0.25 | 1.6 | September | 0.032 | 0.031 | 0.001 | 0.003 | 2019 | This study |
| | Bulan | Shallow freshwater lake | 678 | 0.45 | 7 | January | 0.49 ± 0.04 | 0.91 ± 0.09 | 0.05 ± 0.006 | 0.08 ± 0.07 | 2020–2021 | This study |
| | | | | | | April | 0.09 ± 0.02 | 0.68 ± 0.01 | 0.02 ± 0.0 | 0.002 ± 0.001 | | |
| | | | | | | August | 0.18 ± 0.07 | 0.005 ± 0.001 | 0.06 ± 0.04 | 0.003 ± 0.0 | | |
| | | | | | | September | 0.48 ± 0.11 | 0.18 ± 0.01 | 0.06 ± 0.00 | 0.02 ± 0.01 | | |
| China | Wuliangsuhai | Shallow freshwater lake | 1019 | 305 | 1.5 | Winter | 0.46 ± 0.05 | - | - | - | 2017 | [43] |
| | Keluke | Plateau freshwater lake | 2817 | 54.7 | 5 | Annual average | 0.5 | 0.009 | 0.007 | 0.039 | 1999 | [44] |
| | Hulun | Inland brackish lake | 545.6 | 2339 | 5.75 | October | 0.19 ± 0.04 | - | - | - | 2018–2019 | [45] |
| | | | | | | March | 0.20 ± 0.08 | - | - | - | | |
| | | | | | | May | 0.24 ± 0.13 | - | - | - | | |
| | | | | | | July | 0.16 ± 0.04 | - | - | - | | |
| Russia | Arahley | Shallow freshwater lake | 965 | 54 | 10.2 | Annual average | 0.002–0.004 | 0.25–0.74 | 0.005–0.035 | 0.009–0.014 | 2008–2009 | [23] |
| | Gusinoe | Shallow freshwater lake | 88.3 | 164 | 2–4 | Winter | 0.003 ± 0.003 | 0.09 ± 0.11 | - | 0.006 ± 0.005 | 2016–2018 | [24] |
| | | | | | | Spring | 0.009 ± 0.007 | 0.12 ± 0.08 | - | 0.001 ± 0.002 | | |
| | | | | | | Summer | 0.007 ± 0.006 | 0.09 ± 0.08 | - | 0.003 ± 0.002 | | |

(-) no data.

### 3.3. Seasonal Variation of DIN in Lake Bulan

The percentage of the composition and the amplitude of seasonal fluctuations in the DIN concentration can serve as indicators of the eutrophication of a water body [46]. Average DIN concentration and percentages of $NH_4^+-N$, $NO_3^--N$, and $NO_2^--N$ at the three stations are shown in Figure 6a for the surface water sample, and in Figure 6b for the 7 m depth sample.

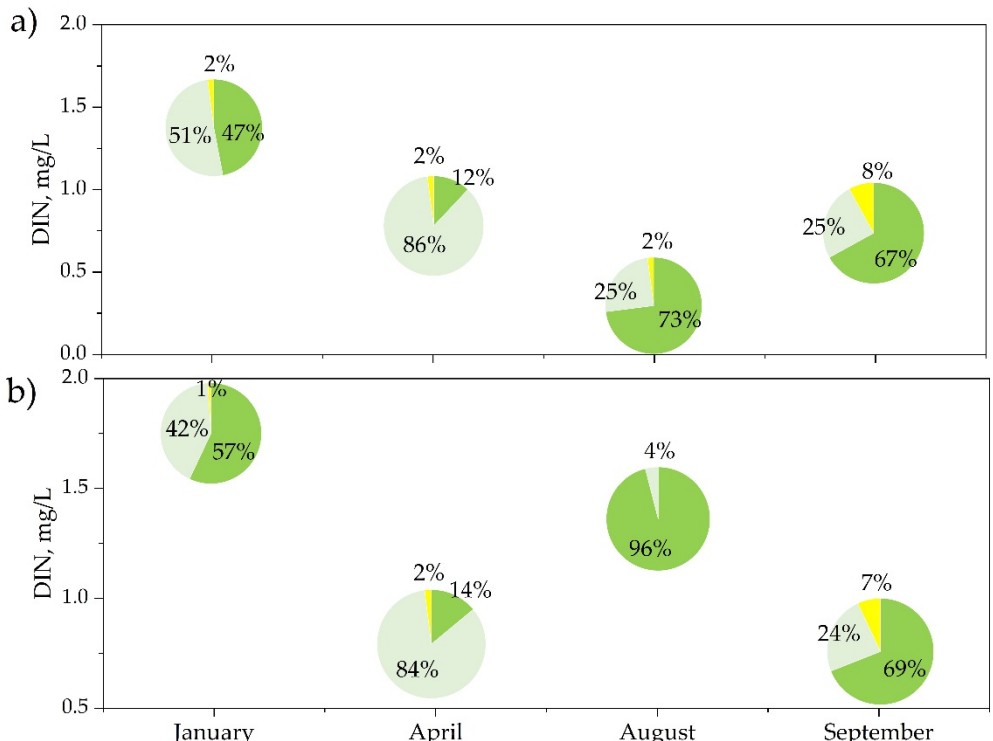

**Figure 6.** Percentages of $NH_4^+-N$ (green), $NO_3^--N$ (light green), and $NO_2^--N$ (yellow) in total DIN concentrations in the water from three sites at Lake Bulan; (**a**) surface and (**b**) bottom layer (7 m depth).

The DIN concentration in the surface water ranged from $0.26 \pm 0.11$ mg/L in August to $1.37 \pm 0.17$ mg/L in January. Higher concentrations of DIN ($1.74 \pm 0.19$ mg/L) were observed in the bottom layer in January. The average percentage of DIN showed clear seasonal differences between surface and 7 m depth water. With eutrophication, the concentration of $NO_3^--N$ increased, as well as its share in the total amount of inorganic nitrogen [46]. The dominant DIN in surface water was $NH_4^+-N$ in August and September, but it changed to $NO_3^--N$ in April. At a depth of 7 m, $NH_4^+-N$ was dominant in August (96%) and lowest in April (14%). The concentrations of nitrites and their share of total nitrogen content were higher in September but decreased to 0–2% in other seasons.

## 4. Discussion
### 4.1. Seasonal Variation of Tropic Status in Lake Bulan

Inorganic phosphates and total phosphorus are used to assess the trophic status of a water body. Phosphorus is a biogenic element used by bacteria and plants in primary production and is often absent from water during the summer. In winter, the phosphate reserve recovers due to phosphorus transition in mineral phosphorus, from dead organisms. The seasonal variation of phosphorus in the surface water samples ranged from $0.03 \pm 0.005$ in August to $0.17 \pm 0.09$ mg/L in January (Supplementary Table S1), while the phosphate concentration ranged from $0.002 \pm 0.001$ in April to $0.085 \pm 0.07$ mg/L in January. In interannual changes, the average ratio of phosphate/total phosphorus (TP) was

0.04 in April, 0.11 in August, and 0.49 in January. This increase provides an indication of the intense biochemical processes in the ecosystem leading to the production of nutrients and decomposition of organic matter in the bottom layer in winter. The lack of DO eventually leads to anoxia, which may impact benthic fish and invertebrates, and increases the internal loading of phosphorus from sediments. The release of phosphate from the benthic sediments in a declining environment has been reported for freshwater lake systems [23]. Some studies have shown that the incidence of $DIN:PO_4^{3-} > 16$ is conditionally conducive to the bloom of Cyanophyta and Chlorophyta [46–48]. Nutrient availability by molar mass ratios of $DIN:PO_4^{3-}$ in Lake Bulan were 117, 1083, 192, and 118 in January, April, August, and September, respectively. Based on molar ratios of $DIN:PO_4^{3-}$, Lake Bulan phytoplankton are limited by phosphorus availability during the year. Low concentrations of phosphorus in North American Lake Michigan pointed to eutrophication from increased anthropogenic phosphate inputs to the lake, which consequently increased diatom uptake of dissolved silica [49]. The lower $SiO_2$ concentrations observed in April in Lake Bulan could be linked to lower concentrations of phosphate and spring bloom of diatoms. This is evidenced by the result of the diatom richness study in the Kharaa River basin [50], which is explained by low temperatures that limit competition from other algal groups and by increased nutrient content due to runoff of snowmelt, soil, and livestock. Increased DOC concentrations often have a significant impact on nutrient cycling due to changes in light attenuation, thermal stratification, and phytoplankton and zooplankton depth distribution caused by its induction. The DOC concentration may also influence the rate of oxygen depletion in lakes through photochemical oxidation, and the availability of critical nutrients such as iron and phosphorus [50,51]. Increased DOC concentrations from $5.82 \pm 0.14$ in April to $9.44 \pm 0.68$ in September result in brown coloration in Lake Bulan, and may be the cause of the relatively low DO concentrations, as they limit UV penetration in the water column [51] (Supplementary Table S1).

*4.2. Dynamic of the DIN Profile in Lake Bulan*

It is notable that the winter and summer profiles of DIN contrast considerably (Figure 7). The $NO_3^- -N$ concentration and vertical profile beneath the ice were similar to those of $NH_4^+ -N$ in winter. This provides evidence of metabolic activity in the lake bottom water. Nitrification is an oxygen consumptive process. High N loading and N cycling have the potential to increase winter depletion rates [16]. The concentration of $NH_4^+ -N$ in bottom water increased over time during summer and winter stratification owing to decomposition of organic matter [21,52,53]. The low DO concentration at a depth of 7 m in August (Figure 4c) can be explained as similar to the thermal classification and summer stratification [10]. Jaeger [54] reported that the lowest DO concentrations were observed during summer in the hypolimnion (6–10 m depth) and surface sediments of a small urban lake, Lake Krupunder in Germany. The depletion of DO is one of the most important phenomena in nutrient load and eutrophication and can directly affect the biogeochemical processes in the water column. Summer and winter stratifications were important regulators of the nutrient cycle in the lake system. Winter nutrient concentrations were higher due to the ice formation on the lake surface and were consumed via primary production periods without ice cover [22]. Bengtsson et al. [55] studied the DO consumption in shallow ice-covered lakes, namely Lake Velen and Lake Vendyurskoe, and argued that the decrease in DO was mainly due to loss in sediments during periods of ice cover, which leads to an increase in the concentration of $NH_4^+ -N$ and $NO_3^- -N$.

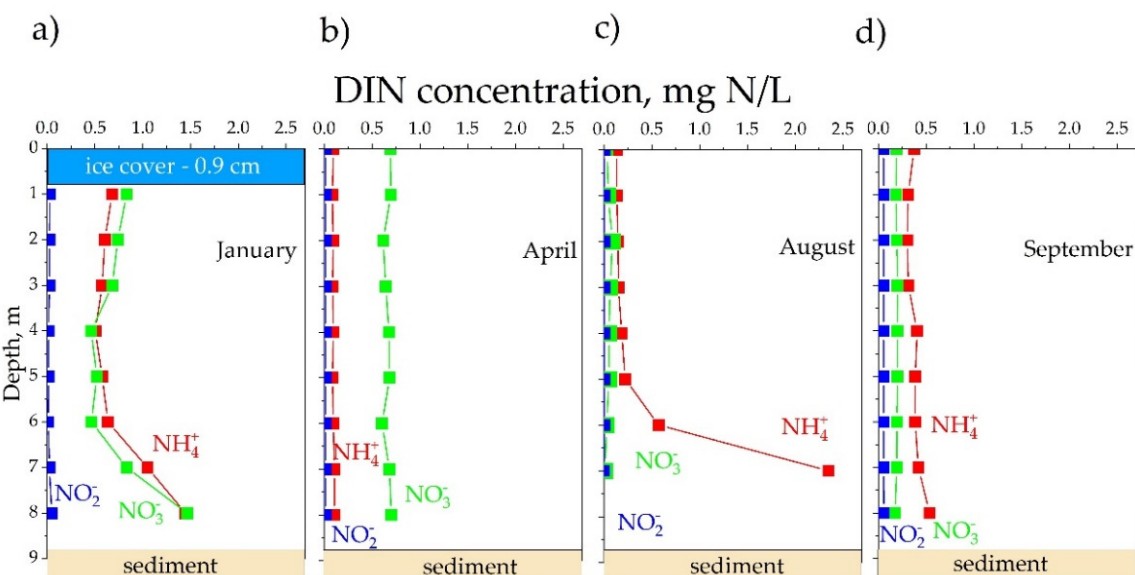

**Figure 7.** DIN concentration profiles at Bulan 2 station in (**a**) winter, (**b**) spring, (**c**) summer, and (**d**) autumn.

The Darkhan city area is characterized as a relatively low winter precipitation region [56]. Lakes in northern Mongolia are covered by ice for four months from November to March [35] and form winter stratification, which leads to the decomposition of organic matter in the bottom layer. With the contribution of snowmelt water, the ice melting process inhibits mixing by wind and increases the WT of the surface layer to 8–10 °C [38]. Lake Bulan experienced the spring change in April, so the concentrations of $NO_3^- - N$ and $NH_4^+ - N$ were mostly similar at different depths, with average values of 0.67 and 0.09 mg/L, respectively (Figure 7b). The decreased concentration of $NH_4^+ - N$ was possibly due to decreased phytoplankton activity. The lower DIN concentration in the surface water during August is indicative of a phytoplankton bloom using $NO_3^-$-N for growth and strong summer stratification, which occurs in the late growth season in the Mongolian Plateau region [57]. On the other hand, a sharp increase in the $NH_4^+ - N$ concentration was observed in the bottom layer, but $NO_3^- - N$ and $NO_2^- - N$ concentrations were reduced (Figure 7c). The concentration of Mn also increased sharply from a water depth of 5 m to the maximum concentration at 7 m, more than 2 m from the benthic sediments (Supplementary Table S1), but dissolved Fe was less than 16 μg/L. As a result, an Mn oxide reducing condition was formed in the bottom layer at a depth exceeding 7 m and/or at the benthic sediment. A decrease in WT from 20.1 °C in August to 14.8 °C in September was observed, and the lake stratification weakened. The vertical distribution of DIN concentrations was mostly constant with depth due to autumn mixing in Lake Bulan, as shown in Figure 7d. Higher DIN concentrations in January were related to low phytoplankton activity and DO used in the nitrification process. Cavaliere et al. [19] described the winter nitrification rate in ice-covered lakes in relation to higher $NH_4^+ - N$ concentrations.

In many eutrophic lakes, oxygen deficiency in the bottom layer occurs annually during summer stratification [15]. The temporal and spatial extent of summer hypolimnetic anoxia is determined by interactions between the lake and its external and internal feedback mechanisms [15]. The knowledge of how these factors have resulted in anoxia in the lake over previous decades will, in part, help to determine the quality of the lake's water in the future.

The predicted acceleration of global warming seems to have significant impacts [9,57] on the thermal conditions of lake water. If air temperature is warmer in late winter and early spring, earlier ice melting [34,58,59] will lead to longer summer stratification and shorter winter stratification. This will possibly increase nutrient concentrations in the

bottom layer and in the water column in the summer season with turnover time in autumn. The shorter winter season may result in a decline in nutrient concentrations in the bottom layer. Therefore, continued monitoring research is needed to quantify the effects of global warming and to identify mitigation measures for these effects.

## 5. Conclusions

Mongolian lake environments have been subjected to the effects of climate change and anthropogenic activity such as urbanization. Lake systems provide livelihood ecosystem services as they are water resources for agriculture and recreational activities. Nutrient concentration and dynamics play an important role in maintaining natural ecosystem diversity. This study focused on a small artificial lake with eutrophication, Lake Bulan, located in the lower reaches of the Kharaa River basin where anthropogenic activity is increasing. Field research was carried out at three sites on Lake Bulan from April 2019 to April 2022. The water samples were analyzed for nutrients, trace metals, and DOC concentrations to understand the present lake environment and to have information for future assessments. The concentrations and vertical profiles of nutrients varied considerably over four seasons. Higher concentrations of DIN and $PO_4^{3-}-P$ in surface lake water were observed in January with values of $1.44 \pm 0.08$ mg/L and $0.08 \pm 0.07$ mg/L, respectively. During winter stratification, the $NH_4^+-N$ and $NO_3^--N$ concentrations increased with depth, with the highest concentrations observed in the bottom layer. During summer stratification, increases in nutrients and Mn concentrations in the bottom layer were observed, with concentrations of 2.35 mg/L, 0.08 mg/L, and 0.37 mg/L for $NH_4^+$-N, $PO_4^{3-}-P$, and Mn, respectively. These results suggest that change in the bottom layer and benthic sediments due to organic matter decomposition during the summer and winter seasons is evidence of a declining environment. Water stratification is one of the factors controlling nutrient concentration, which is related to phytoplankton activity in Lake Bulan. Future detailed studies will be necessary to understand the effects of global warming and anthropogenic activity on small lakes in Mongolia, among which are periods of shorter winter stratification and longer summer stratification.

**Supplementary Materials:** The following supporting information can be downloaded at: https://www.mdpi.com/article/10.3390/w14121916/s1, Table S1: Dynamics of inorganic nutrients, cations and trace metals in water Lake Bulan.

**Author Contributions:** A.B. and S.N. wrote the paper and analyzed the data. A.B., S.T., K.F. and B.G. performed the experiments. A.O., E.M., B.G., B.Z., A.Y. and D.D. carried out the fieldwork. All authors have read and agreed to the published version of the manuscript.

**Funding:** Financial support was provided by the Higher Engineering Education Development Project; Functional material based on Mongolian Natural Minerals for Environmental Engineering, Cementitious and Float Process (No. J11A15), under collaborative research (20012 and 21002) of the Joint Usage/Research Program of Kanazawa University and the Japanese Society for the Promotion of Science (JP18KK0296 and JPJSBP120219915).

**Institutional Review Board Statement:** Not applicable.

**Informed Consent Statement:** Not applicable.

**Data Availability Statement:** Not applicable.

**Acknowledgments:** The authors thank T. Yoshimura and Y. Yamashita of Hokkaido University and R. Sugimoto of Fukui Prefectural University for their assistance with nutrient and DOC analysis.

**Conflicts of Interest:** The authors declare no conflict of interest.

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
