# Peer review of "Seasonal Variation and Vertical Distribution of Inorganic Nutrients in a Small Artificial Lake, Lake Bulan, in Mongolia"

_water, doi:10.3390/w14121916_

Round 1

Reviewer 1 Report

The authors have addressed all the issues in the comments, and I suggest that this paper be accepted without further revision.

Author Response

Dear Reviewer,

Thank your for your interest in our manuscript. We appreciate your careful review.  We are sending to you the last revised version of our manuscript. We improved the figures to more clear visiation. 

Reviewer 2 Report

This is a resubmited manuscript. Although the authous didnot have a answer for the last comments, the manuscript did improved tremdously for several aspects. First, a more detailed and profound analysis of water stablity by using Gibbs plot was performed which is more interesting than the raw data as the former manuscript. Second, unrelevent samples and data was removed, such as sewage treatment effluent, which makes the aim of manuscript more clear and logically reasonable.Third, the data and figure, especially for the vertical profile under different season, are improved greatly and easy to read. This is quite important to make it differ to a scientific report. Fouth, the data quality, like significant digits and QA was much improved. So I like  this manuscirpt at the present form.

To improve the quality of this manuscript, I would suggest

1. to improve the Fig.6. By using diffent area of circle as the concentration of DIN and put the 4 seasons into a X(season) scale and total concentration of DIN as a Y scale. This would be useful for vivid presentation of the results.

2. Discussion 4.1. In this part, the results for DIN, DIP should be a results part. and the ratio of NP and etc should be in the discussion part

3. For discussion 4.2, if phytoplakton or Chla was present, I would encourage a disscussion for the relationship between nutrients, ratios and Primary Producers. This work would make it more important to address the nutrients dynamic and its ecology effect.

4. In the methods part, Fe, Mn was measured, but I didnot see any results and discussions in the contents. So is this necessary to mention you also measured the Fe and Mn. 

Generally, this verison has tremdously improved and I thanks the big work for this.

Author Response

Dear Reviewer,

Thank your for your interest in our manuscript. We appreciate your careful review and constructive suggestions  to improve the quality of our manuscript.  Following this letter are the comments given by reviewer with our responses including where the necessary changes were made line by line in the manuscript which are marked with yellow lightened in the main body of the manuscript. 

Reviewer 3 Report

Dear Authors,

after reading your manuscript have only a few comments that are listed below. Besides those, I consider the paper positively for publication in Water. I can confirm that the paper would have high readers interest, it is well developed and presents results from an interesting in terms of environmental sciences and from not very accessible part of the world.

Specific comments:

Line 58 & 62: Small lakes - do you mean shallow lakes? Its not the same?

Line 227: In Figure 4 and 5 I would add a normal legend with images of triangles/dots/squeres.

Moreover, please try to increase the size of each figure in fig 4 while it is difficult to see each result. 

Author Response

Dear Reviewer,

Thank your for your interest in our manuscript. We appreciate your careful review and constructive suggestions  to improve the quality of our manuscript.  Following this letter are the comments given by reviewer with our responses including where the necessary changes were made line by line in the manuscript which are marked with yellow lightened in the main body of the manuscript.

This manuscript is a resubmission of an earlier submission. The following is a list of the peer review reports and author responses from that submission.

Round 1

Reviewer 1 Report

Baterdene and co-authors present a study about the seasonal variation and vertical distribution of nutrients in a small artificial lake in Mongolia. The topic is surely interesting as water bodies worldwide suffer from multiple anthropogenic impacts, related to both eutrophication and climate change. However, I cannot express positively. As it stands, the paper reads more like a regional technical report about the water quality evaluation of the Bulan Lake. I think there needs to substantially rewrite the introduction, discussion, and conclusions to convert the manuscript into a scientific paper potentially interesting for an international audience. At present, the entire Introduction is focused on environmental problems of the study area, the Kharaa River basin. The same considerations is valid for the Discussion. Furthermore, a deeper and speculative interpretation of the presented data is needed, in a way to better evaluate both the scientific level of the presented experimental approach and also their spendibility for practical application in management programmes of human-impacted watersheds worldwide and not limited to Mongolia.

Reviewer 2 Report

Paper “ Seasonal variation and vertical distribution of inorganic nutrients in a small artificial lake, Bulan Lake in Mongolia” is of obvious data report type. First, the paper name is of Bulan Lake, but in the context of this paper, River Kharaa and two other lakes (Bornuur and Ugii) are also included. Again, even WWTP and its effect to the river nutrients was also described. So it is very vogue of this paper to show the actual scientific merit. Second, this paper is poorly constructed and read like a report. I would suggest to remove all the irrelevent data like the river and WWTP problem. Third, this paper is of empirical results. For instance, the lake stratification is an important factor when talking about the nutrients vertical profiles. But the stratification (eppilimnion and hypolimnion) is divided without any temperature profile evidence. Fourth, TOC, Nitrogen, phosphorus and major cation are displayed without clear scientific purpose. Lastly, the authors showed this paper will look on the global warming and human activity, but these two driving forces are pooly quantified and the relationship between them to the nutrients is hardly analysis quantitatively.

Some minor problem listed for future revisement:

L21-23 these sentences should be reconstructed. Please note the dynamics of nutrients are not just nutrients profile. It is needed to elucidate what the dynamics exact mean.

L35-36 it is hardly to indicate the importance of stratification on the biochemical cycling.

L41-42 how to prove the nutrient concentration level were related to global warming and eutrophication by human activity

Table 2.3.4 is not suitable as this format

For table 3 I recommend to use Gibbs plot to document the water source.

Table 4 very interesting. Some data of Kharaa 4 is even lower than the upperstream Kharaa 2. Could you explain why?

Figure 2. is the data of this figure equivalent to the table 4? Is so this would be repetitive results

L188-L222 is very tedious. Better to make it more concise.

L123, here showed there are twice of sampling for winter, but I can’t find the data

L 147 for should be within

L291 this part are only for rivers. And the lake are not connect with this river, so why discuss this river?

Table 5. There are tremendous data for lake nutrients. This comparison is of very little importance. If selection of these lakes are of some criteria, please clarify.

L354 this equation are poorly use only for lakes in temperate regions. Please check if it is suitable for use for these investigated lakes.

L374-387 are not concise enough. Better to rewrite.

Reviewer 3 Report

In this paper, water samples from a small artificial shallow lake (Bulan Lake) and the Kharaa river were collected seasonally from April 2019 to September 2021. The authors analyzed nutrients, major cations, trace metals, and dissolved organic carbon (DOC) to understand Bulan Lake's seasonal variation and vertical distribution of inorganic nutrients. The results of this paper can provide a theoretical basis for recent environmental issues in Mongolia and provide some reference for other scholars studying nutrient problems in lake systems.

  1. Lines 124-125 of the text state: Surface and bottom water samples were collected from two stations in Bulan Lake (Bulan1 and Bulan3), but there is only one depth at both sites in Table 1. How can surface and bottom water samples be represented? It is recommended that all water depths are shown in the Table
  2. Line 169 says: "DO concentration ranged from 0.33 to 4.72 mg/l", how did 0.33 come about? This data is not in the Table, and please verify.
  3. Why are "OR" and "Transparency by Secchi desk, m" listed in Table 2, and is it useful in the text?
  4. In Table 2, "Transparency by Secchi desk, m," why only one time data, according to the normal summer Transparency should be able to measure? Please double-check.
  5. lines 174-175 read: " Kharaa River water had a high Na+ +K+ concentration in the Darkhan city area." Please specify which station corresponds to "the Darkhan city area."

6.Lines 178-179 state: "Kharaa River water mainly contained cationic ions in the order of Ca2+ > Mg2+ > Na+ +K+" How was this conclusion reached? Please explain

  1. Why are there only three seasons of data in Table 3? Please explain.
  2. Please double-check that the data appear in rows 188-211 correspond to the data in the Table.
  3. Lines 207-210 present: " The concentration of NH4+ exceeded the national standard of water quality 6.6-fold and 23.6-fold and was also higher than average for the country by 3.3-fold and 11.7-fold in April and January at station Kharaa 3, which was located downriver from the WWTP." It is suggested to list "the national standard of water quality" and "average for the country" to facilitate the comparison.
  4. Lines 255-256 read: " The NO2 concentration showed a wide variation from 0.01 mg N/l in January to 0.057 mg N/l in September " Please check if the expression in the text matches the figure.
  5. lines 274-275 state, "The seasonal values of phosphorus ranged from 0.028 to 0.288 mg/l (Supplement Table 2)". It is recommended that the values expressed in the text be consistent with the numbers in the Table. It is recommended that the values expressed in the text be compatible with the numbers in the Table in decimal places.